# Coordination and variability of muscular activation in male athletes with and without subacromial impingement syndrome: A case-control study

**Rahman Sheikhhoseini** [1]*, **Sajjad Abdollahi**[2], **Mohammad Salsali**[3], **Mehrdad Anbarian**[4], **Trent M. Guess**[5]

1 Department of Corrective Exercise & Sport Injury, Faculty of Physical Education and Sport Sciences, Allameh Tabataba'i University, Tehran, Iran, 2 Department of Sport Biomechanics, Faculty of Sports Sciences, Bu-Ali Sina University, Hamedan, Iran, 3 Faculty of Physical Education and Sport Sciences, Allameh Tabataba'i University, Tehran, Iran, 4 Department of Sport Biomechanics, Faculty of Sports Sciences, Bu-Ali Sina University, Hamedan, Iran, 5 Department of Physical Therapy, University of Missouri, Columbia, Missouri, The United States of America

* Rahman.pt82@gmail.com

## Abstract

### Objectives

Athletes with Subacromial Impingement Syndrome (SIS) exhibit altered muscular coordination and variability during repetitive shoulder movements compared to athletes without SIS. This research compared the Coordination and Variability of Muscular Activation in Male Athletes with and Without SIS.

### Methods

In this case-control study, twenty-four male athletes were recruited and divided into two groups: those with SIS (n = 12) and those without SIS (n = 12). Participants performed a repetitive reaching task (RRT) for a minimum of fifteen repetitions. Electromyography (EMG) data were recorded from selected shoulder muscles. Muscle synergies, intra-group variability, and inter-group variability were extracted from the EMG data. An independent sample t-test or Mann-Whitney U-test was employed to analyze data at a significance level of 95% (α < 0.05).

### Results

Three observable muscle synergy patterns were identified in both groups. Significant differences in variance accounted for (VAFmuscle) were found in the posterior deltoid, subscapular, and middle deltoid muscles, but these differences may not be clinically significant and warrant further research. No significant statistical differences were found in intra-group variability between the groups, which may suggest that the hypothesis is not fully supported. However, significant differences in inter-group variability were observed between the SIS and control (CON) groups.

**Data availability statement:** All relevant data are available in the paper and its Supporting Information files.

**Funding:** Iran National Science Foundation (INSF) under project no. 4013596.

## Conclusions

This study showed differences in muscular coordination and variability during RRT in athletes with and without SIS. Three different muscle synergy patterns were demonstrated in both groups. It seems that timing and coordination changes in muscle activation may influence movement efficiency and increase the risk of performance errors.

## Introduction

Shoulder pain is a common issue among athletes. Shoulder pain is often reported in individuals who repetitively use their arms during sports activities and leisure time [1–10]. In sports involving overhead movements, shoulder pain might develop into chronic and disabling conditions. Repetitive movements in the upper body constitute a significant part of sports and leisure activities [11–16] and are usually accompanied by discomfort and pain in the shoulder region [11].

Performing repetitive movements during sports may predispose athletes to Repetitive Movement Disorders (RMD) [17]. However, most research investigated RMDs in the lower back region, leading to insufficient information about the causes of RMD in the upper limbs [18]. Various biomechanical studies have been conducted to understand factors related to Subacromial Impingement Syndrome (SIS) in athletes [19]. Chronic pain is accompanied by reduced movement speed [20] and increased electromyography (EMG) activity in the shoulder stabilizer muscles, aiming to preserve motor function and minimize the activity of global muscles in painful areas [21]. However, few studies have addressed coordination and variability among athletes with SIS during repetitive movements.

Muscle coordination is a control strategy for creating an efficient, smooth, and precise movement pattern. It is defined as the ability to organize appropriate relationships between active muscles during a motion [22–24]. Coordination variability demonstrates adaptability in different movement patterns in response to internal and external disturbances [2,3,22–24]. Therefore, muscular coordination seems to be an appropriate indicator of control mechanisms during repetitive activities [4].

According to the current literature, muscle synergies signify a more advanced organization of motor control, concentrating on the coordinated function of muscle groups during movement [25]. By examining muscle synergies, researchers can reveal how individuals utilize neuromuscular strategies to perform tasks effectively, adjust to discomfort, or accommodate functional limitations. This method offers a broader understanding of how the central nervous system coordinates movement [26,27].

As mentioned, SIS is a common injury among athletes, especially those involved in overhead movements [28,29]. Chronicity of this condition is more probable in athletes requiring such movements [30]. Furthermore, SIS can decrease athlete performance, incurring costs for athletes, clubs, or healthcare systems [28]. Therefore, numerous studies have been conducted to understand potential factors contributing to SIS among athletes. Additionally, various studies indicate that muscular coordination and variability can reflect the appropriate function of movement structures in the body [31]. In this regard, it's shown that coordination and variability among muscles change in the presence of chronic pain [32]. However, no study has specifically addressed muscular coordination and variability during repetitive shoulder movements among athletes with SIS. Hence, this research aims to investigate and compare muscular coordination and variability during repetitive shoulder movements among athletes with and without SIS.

## Methods

### Participants

In this case-controlled study, 24 male athletes were recruited with the convenience sampling method and divided into two groups of athletes with SIS (12 persons) and without SIS (12 persons). Sample size estimation showed that a minimum of 22 male athlete volunteers would be necessary to achieve 90% statistical power at an alpha level of 0.05 within the two groups. According to a previous study, the sample size estimation was performed based on a calculated effect size (1.66) (extracted by G*Power ver 3.1 software) when investigating the effect of repetitive movement tasks on shoulder muscle activation on lower trapezius [33]. This study included male athletes aged 20–35 with at least four years of training experience in overhead sports among groups with and without SIS.

For the SIS group, the athletes should have had a history of lateral shoulder pain for a minimum of 3 months on their dominant hand that the athletes usually used to throw a ball, had no pain radiations to the cervical or distal segments of the upper extremity, the pain should have aggravated with playing or training and relief with rest, and had no history of fever or night pain and cold sweat during the pain period. Additionally, clinical tests for Subacromial Impingement, such as the Neer Test and Hawkins-Kennedy Test, were performed to confirm the diagnosis of subacromial impingement syndrome. The exclusion criteria were as follows: athletes with a history of cervical radiculopathy, glenoid labrum lesion, shoulder dislocation, and shoulder girdle muscle rupture, the history of neurological, vestibular, or other balance-affecting medical conditions, and using any medications for neurological or metabolic disorders. An expert-certified musculoskeletal physical therapist performed the inclusion and exclusion criteria and diagnoses of shoulder pain with 16 years of experience in the field. Before starting the investigation, study approval was obtained from the Biomedical Research Ethics Committee of Allameh Tabataba'i University (ATU) (Ethics code: IR.ATU.REC.1401.084), and all participants gave written informed consent. The authors confirm that all research was performed in accordance with relevant guidelines/regulations. All subjects were right-handed. Sampling in the laboratory phase of this study began on September 23, 2023, and continued until April 13, 2024.

### Experimental protocol

The athletes were asked to perform a repetitive reaching task (RRT) for a minimum of fifteen repetitions. The last ten attempts were used as laboratory data for analysis. During the task, athletes were asked to stand on two adjacent force plates with their feet positioned relaxed at shoulder-width apart (Fig 1). A researcher-made machine was used to control the arm movement range and maintain the upper extremity's testing position during the RRT. The testing shoulder was placed in the scapular plane at 90 degrees of elevation. This machine had two adjustable cylindrical arms; one was matched with 30% and the other to 100% of the testing upper extremity length. Moreover, a circular mesh barrier was adjusted under the elbow to keep a consistent arm position during the RRT. The athletes were not allowed to touch this plate during the RRT. Then, the athletes were requested to continuously perform elbow flexion/extension so that the index finger could touch the 30% and 100% targets matched with the machine arms while the arm and elbow were maintained in the predetermined position. A metronome with auditory feedback was used to keep a rhythm of one trial per second so that the athletes could touch each target without hearing a sound. During the RRT, electromyography (EMG) data were recorded, too. Subjects continued the RRT until they had 15 successful elbow flexion/extension movements. Importantly, participants were unaware of the stopping movement.

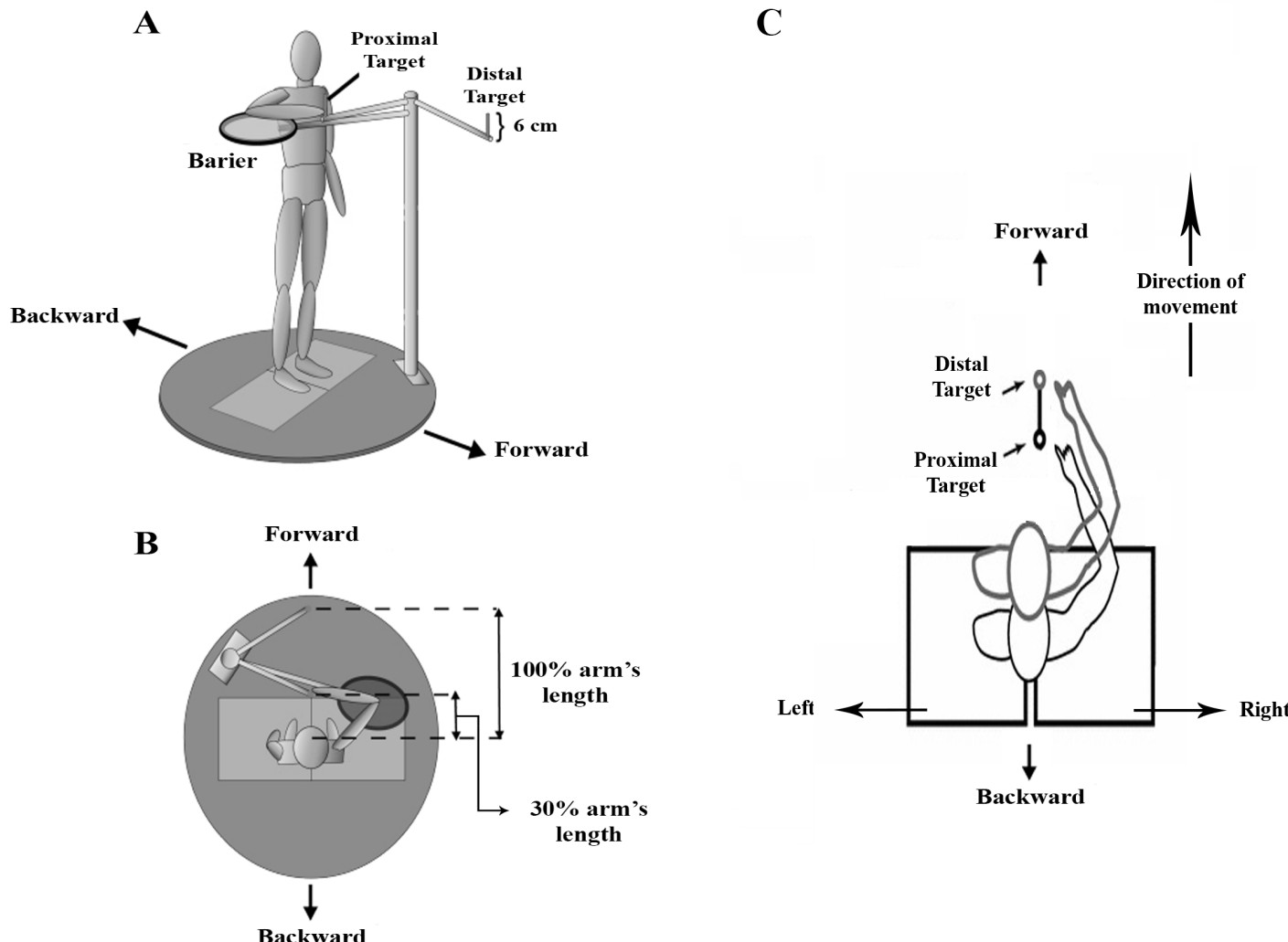

**Fig 1. The laboratory setup for the repetitive movement task machine is showed from the front-right (A) and top (B) views.**

## Data acquisition

After preparing the skin, disposable Ag/AgCl chloride electrodes were attached to the muscles in accordance with previous recommendations [34]. Upper trapezius (TS upper), infraspinatus (IS), biceps brachii (BS), anterior deltoid (AD), lower trapezius (TS lower), posterior deltoid (PD), middle deltoid (MD), and triceps medial head (TB) EMG data were obtained from the examination side based on the previous guidelines [35]. Before starting the main test, a maximum voluntary isometric contraction test (MVIC) was taken from all the above muscles based on the recommended standards [36].

A TeleMyo 900 EMG measurement device (Noraxon USA Inc., Arizona, USA) was used to record the EMG data. A 16-bit A/D card was used to digitally transform the 1200 Hz sampled data, which was stored for later examination. The participants were told to continue touching and going during each repetition, alternating between forward and backward phases. The arm and finger movement were used to define the movement phases. The forward phase began when the fingers made contact with the proximal target, and the end position represented the

distal target of the same phase. The reverse phase was defined by using the identical occurrences in the opposite way.

## Data analysis

For the pre-processing of the EMG data, firstly, we filtered using a band-pass filter (20–450 Hz), and then, we rectified and smoothed with a low-pass filter (12 Hz, 4th order Butterworth) for each signal. Furthermore, the signals were standardized to 90% of the maximal voluntary isometric contraction, with an average value of 100 ms across the peak of the linear envelope of the set of three repeats. Lastly, we interpolated each linear envelope to 100 points, as this study's goal was not to quantify the rate of muscle activation [37]. Since different people may experience the same relative time phases differently, a time normalization from 0% to 100% was utilized to relativize each phase for the forward and backward phases of the RRT movement (a flowchart illustrating these steps is provided below (Fig 2)).

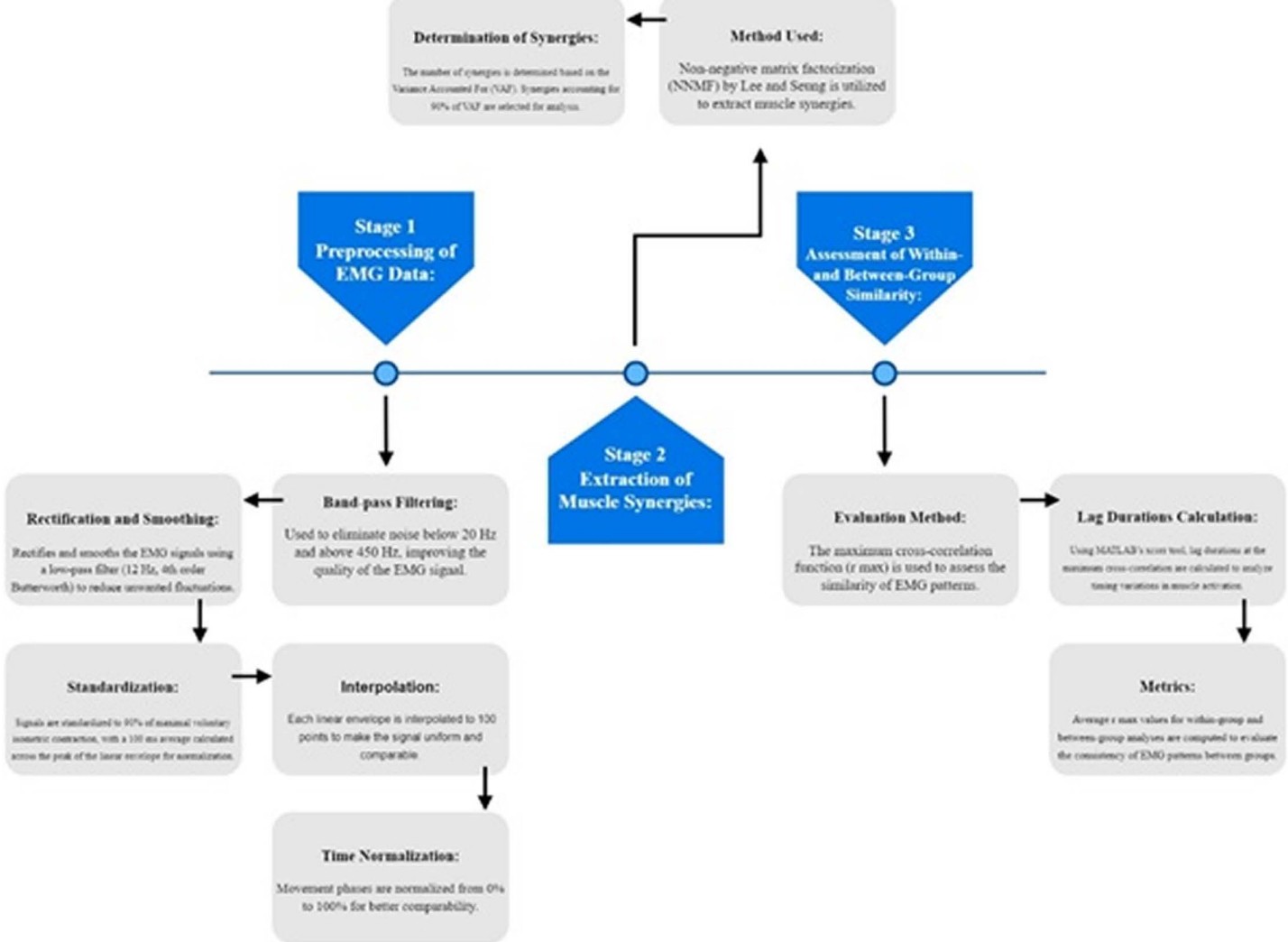

**Fig 2. Flowchart illustrating the steps involved in the pre-processing and analysis of EMG data.**

## Extraction of muscle synergies

Lee and Seung's non-negative matrix factorization (NNMF) approach was utilized to extract muscle synergies. The normalized linear envelope of EMG signals was decomposed using NNMF. This approach obtains the decomposition of the starting matrix by minimizing the residual Frobenius norm between the matrices. The methods were based on the equations provided in a previously published paper [38].

In this study, we include a figure that illustrates an example of a normalized EMG linear envelope, the three Synergy Excitation Primitives obtained from the NNMF analysis, and the EMG signal reconstructed from these components.

In addition to the NNMF applied to extract muscle synergies, the follow-on analyses involve several key steps to evaluate the synergy activation coefficients and assess the temporal consistency of muscle activity patterns. After the NNMF decomposition, we calculated the within- and between-group similarity of synergy patterns using the maximum cross-correlation function (r-max), which measures the waveform similarity across participants. To evaluate the robustness of muscle synergies, we also computed the variance accounted for (VAF) for each muscle, ensuring that the derived synergies adequately captured the underlying muscle activation patterns. The variance accounted for method was used to determine the number of synergies to retain. For further details on these analyses, readers can refer to the methods described in our previous publications.

This study determined the number of muscular synergies by the VAF. In this line, the number of muscular synergies that characterized 90% of VAF was considered the number of synergies extracted for analysis. Since the algorithm does not employ the same sorting method for every set of participants, the obtained muscle synergies were sorted manually.

## Assessment of within- and between-group similarity

Additionally, differences in synergy activation coefficients and individual EMG patterns were evaluated using the r-max, a measure of waveform similarity. To assess the quality of this match, we utilized the VAF as a key metric, with a VAF value above 90% indicating an acceptable match. Additionally, we assessed the within- and between-group similarity of synergy patterns using the r-max, which measures waveform consistency across participants.

To evaluate variations in the timing of muscle and synergy activation, we concurrently calculated the lag durations at the maximum cross-correlation function using Matlab 2019a (Mathworks Inc., Natick, MA, USA) xcorr tool for centered data (option "coeff"). By averaging the r max-values of each within and between-group pairwise analysis, the within and between-group index of variability was determined. The determined metrics served as markers for the waveform consistency both between and within the following groups: 144 pairs with and without SIS (12 healthy × 12 with SIS), 66 pairs with SIS (each participant compared with the other 11), and 66 pairs with SIS (each participant compared with the other 11). Furthermore, for every participant pair, the R-values of the muscle synergy vectors were computed [39,40].

The start and end points of the motion in the EMG signal were determined using a synchronized signal from the researcher-made device that tracked the arm movement. This device provided precise markers for the onset and completion of each movement cycle, which were then aligned with the corresponding EMG recordings. By synchronizing the EMG signals with the motion data, we ensured accurate data segmentation prior to time normalization.

To assess the degree of muscle synergy similarity between athletes with and without SIS, we evaluated whether the muscle synergies extracted from each participant explained the EMG

patterns of both groups of athletes. The methods used were based on the equations provided in a previously published paper [38].

## Statistical analyses

The Shapiro-Wilk test was utilized to check the data's normal distribution. Based on the normality check results, an independent sample t-test or Mann-Whitney U-test was employed to analyze data at a significance level of 95% ($\alpha < 0.05$) (by using SPSS software, version 25.0, SPSS Inc., Chicago, IL, USA). Partial Etta squared was used as effect size (ES).

# Results

## Characteristics of the participants

Table 1 summarizes the demographic data for the study participants. Notably, there were no significant differences in age, height, body mass, and BMI between the two groups.

## Muscle synergies

Applying the criteria for identifying muscle synergies, three muscle synergies exist in both the CON and SIS groups. To clarify, the NNMF was conducted simultaneously on the combined EMG data from both groups, thus facilitating a comprehensive analysis of the extracted synergies. These extracted synergies contributed to a comparable VAF between the two groups: 0.90 ± 0.06 for CON participants and 0.89 ± 0.12 for SIS, as represented in Table 2. Importantly, no statistically significant differences in VAF were detected for the three extracted muscle synergies between the two groups ($p = 0.841$; ES = 0.002).

For both groups, the AD muscle accounted for more than 75% of the VAF, while the PD and MD muscles were significant contributors in the SIS group. Differences between groups were found in $\text{VAF}_{muscle}$ of PD ($p = 0.016$; ES = 0.236), MD ($p = 0.022$; ES = 0.215), and TB ($p = 0.006$; ES = 0.299) muscles. The VAF of each muscle is presented in Table 3 for enhanced clarity.

In the CON group, muscle synergy #1 primarily corresponds to the posterior muscles of the upper limb (TS upper, TS lower, PD, IS, TB), which are predominantly active during the reverse phase of the movement. Muscle synergy #2 predominantly engages the anterior muscles of the upper limb (BB, AD, MD), which are active during the forward phase and, to a lesser extent, during the initiation of the reverse phase. Muscle synergy #3 represents the conclusion of the forward phase and involves mainly the arm muscles (AD, MD, BB, TB, TS upper, TS lower), which are active from the middle of the forward phase to the onset of

**Table 1. Participant demographics.**

| Variable | CON (N = 12) | SIS (N = 12) | P-value |
|---|---|---|---|
| | Mean ± SD | Mean ± SD | |
| Age (years) | 21.66 ± 2.01 | 23.25 ± 2.76 | 0.123 |
| Height(cm) | 183.33 ± 12.82 | 182.58 ± 6.90 | 0.860 |
| Body Mass (kg) | 78.50 ± 12.70 | 72.66 ± 8.74 | 0.204 |
| BMI (kg.m$^2$) | 23.33 ± 2.78 | 21.77 ± 2.08 | 0.134 |
| VAS (mm) | N/A | 2.83 ± 0.83 | – |

**Notes:** †Based on the results, no significant differences were observed between groups. Statistically significant differences between the CON and SIS groups. **P-value 0.05 considered to be statistically significant.

Abbreviations: CON = Control Group; SIS = Subacromial Impingement Syndrome Group; BMI = Body Mass Index; VAS = Visual Analogue Scale; SD = Standard deviation; Cm = Centimeters; Kg = Kilograms; Kg.m2 = Kilogram per square meter; Mm = Millimetre.

**Table 2. Mean ± SD values of variance accounted for (VAF) relative to the original extraction iteration of muscle synergies for the Control Group (CON) and Subacromial Impingement Syndrome Group (SIS) for the repetitive reaching task (RRT).**

| Variable | | CON (N = 12) | SIS (N = 12) | P-value[†] | ES |
|---|---|---|---|---|---|
| | | Mean ± SD | Mean ± SD | | |
| VAF (%) | #1 | 0.82 ± 0.75 | 0.81 ± 0.17 | 0.786 | 0.003 |
| | #2 | 0.90 ± 0.10 | 0.89 ± 0.14 | 0.910 | 0.003 |
| | #3 | 0.92 ± 0.09 | 0.91 ± 0.12 | 0.893 | 0.001 |
| | #4 | 0.95 ± 0.06 | 0.95 ± 0.07 | 0.897 | 0.001 |
| VAF$_{total}$ (%) | | 0.90 ± 0.06 | 0.89 ± 0.12 | 0.841 | 0.002 |

**Notes:** [†]statistically significant differences between the CON and SIS groups. **P-value,0.05 considered to be statistically significant.

Abbreviations: #1, Synergy #1; #2, Synergy #2; #3, Synergy #3; ES, effect size.

**Table 3. Mean ± SD values of variance accounted for each muscle (VAF muscle) regarding a three-synergy model for the repetitive reaching task (RRT).**

| Variable | VAF$_{muscles}$ | | | |
|---|---|---|---|---|
| | CON (N = 12) | SIS (N = 12) | P-value[†] | ES |
| | Mean ± SD | Mean ± SD | | |
| AD | 0.75 ± 0.18 | 0.81 ± 0.17 | 0.470 | 0.024 |
| PD | 0.69 ± 0.26 | 0.92 ± 0.14 | 0.016** | 0.236 |
| MD | 0.70 ± 0.28 | 0.92 ± 0.13 | 0.022** | 0.215 |
| IS | 0.67 ± 0.27 | 0.60 ± 0.37 | 0.598 | 0.013 |
| TS$_{upper}$ | 0.50 ± 0.29 | 0.59 ± 0.32 | 0.492 | 0.022 |
| TS$_{lower}$ | 0.50 ± 0.25 | 0.61 ± 0.35 | 0.395 | 0.033 |
| BS | 0.56 ± 0.32 | 0.44 ± 0.25 | 0.322 | 0.045 |
| TB | 0.62 ± 0.15 | 0.36 ± 0.24 | 0.006** | 0.299 |

**Notes:** [†]statistically significant differences between the CON = Control Group; SIS = Subacromial Impingement Syndrome Group.

**P-value 0.05 is considered to be statistically significant.

Abbreviations: TS$_{upper}$: upper trapezius, TS$_{lower}$: lower trapezius, IS: infraspinatus, AD: anterior deltoid, MD: middle deltoid, PD: posterior deltoid, BS, biceps brachii, TB, triceps medial head; ES, effect size.

the reverse phase. In the SIS group, while the overall composition of each muscle synergy was similar, there were notable differences. In synergy #1, the IS and TB muscles exhibited a reduced weighting in the SIS group. Furthermore, synergy #2 demonstrated a similar trend, showing that the BB also had a diminished weighting. Finally, in synergy #3, both BB and TB muscles presented a lower weighting in the SIS group.

## Intra-group variability

Concerning the synchrony of muscle synergies (i.e., the time lag), no significant shifts were observed in either the CON or SIS groups. Furthermore, all maximum correlation values (r max) exhibited a robust correlation, ranging from 0.70 to 0.87. Similarly, the muscle synergy vectors also demonstrated strong correlations in both groups, ranging from 0.73 to 0.87.

In both groups, individual EMG patterns exhibited strong correlations (0.75 < r max < 0.88). However, for SIS, shifts were found in UT (p = 0.003; ES = 0.323) and PD (p = 0.001; ES = 0.392), and for CON, shifts were also found in UT (p = 0.01; ES = 0.341), LT (p = 0.001; ES = 0.249), MD (p = 0.003; ES = 0.255), and TB (p = 0.002; ES = 0.228).

## Inter-group variability

When comparing the intra-group variability of both groups, for synergy #2, there were no significant differences in r-values and r-max-values between muscle synergy vectors and synergy activation coefficients. However, for synergy #1 and synergy #3, a significant difference in r max-values was observed. Specifically, for synergy #1, the CON group exhibited a higher correlation value (p = 0.001; ES = 0.234). Similarly, for synergy #3, the CON group showed a higher correlation value (p = 0.001; ES = 0.241), as presented in Fig 3.

In the inter-group analysis, significant shifts were found in the three extracted synergies, synergy #1 activated earlier for the SIS group (p < 0.001; ES = 0.167). Synergy #2 was also activated earlier for the SIS group (p < 0.001; ES = 0.284). Conversely, synergy #3 activated earlier for the CON group (p = 0.005; ES = 0.342), as presented in Fig 4.

The synergy activation coefficients exhibited strong correlations, with similarity indexes of 0.21, 0.20, and 0.19 for synergies #1, #2, and #3, respectively. However, the muscle synergy

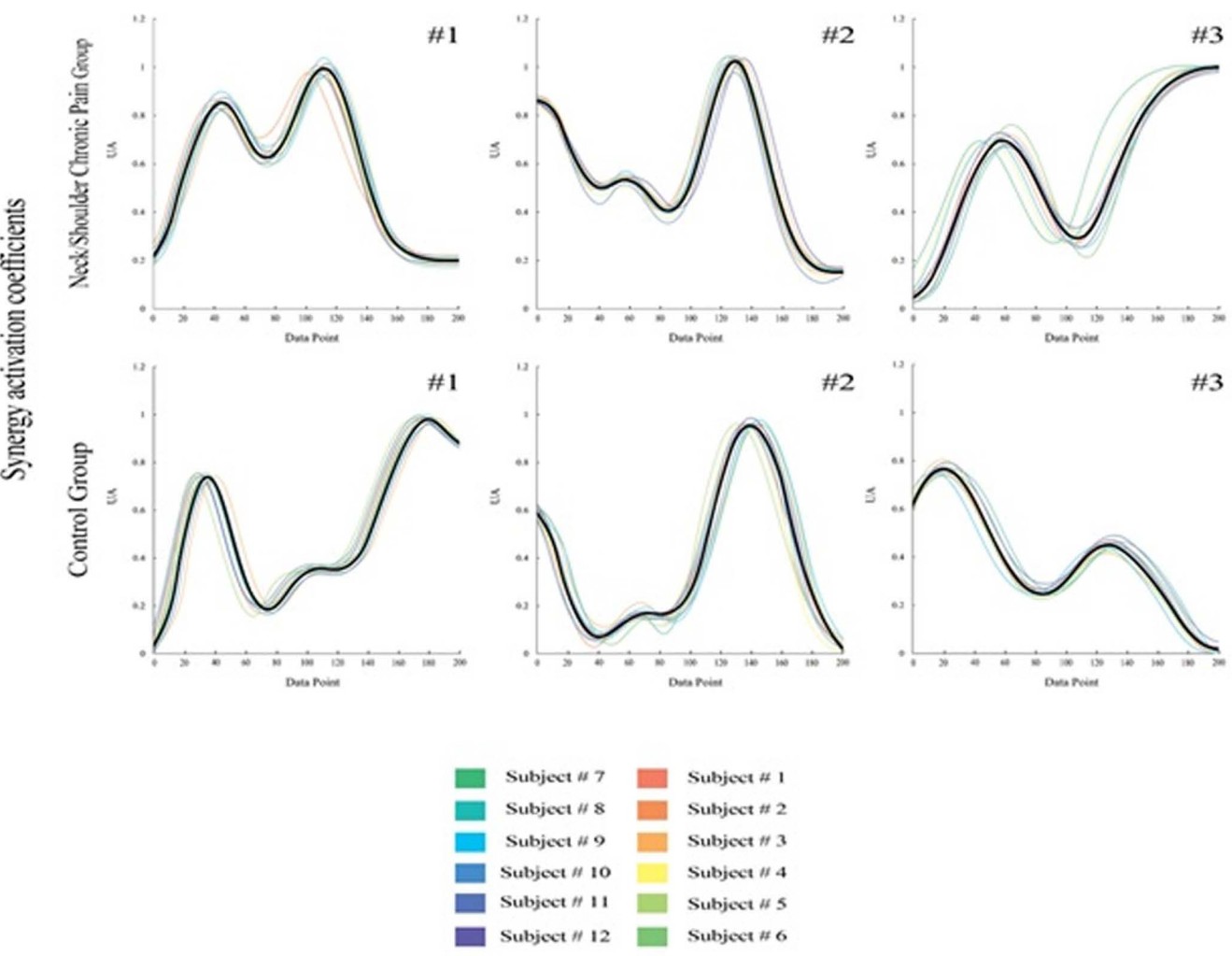

**Fig 3. Averaged Inter-individual variability of Synergy activation coefficients for all the subjects of Control Group (CON) and Subacromial Impingement Syndrome Group (SIS) (UA—arbitrary units).** *Top panel*: corresponds to SIS while *Bottom panel*: regards to CON. The thick black line represents the group mean, while the thin lines represent individual synergy activation coefficients. (#1: synergy 1, #2: synergy 2, #3: synergy 3).

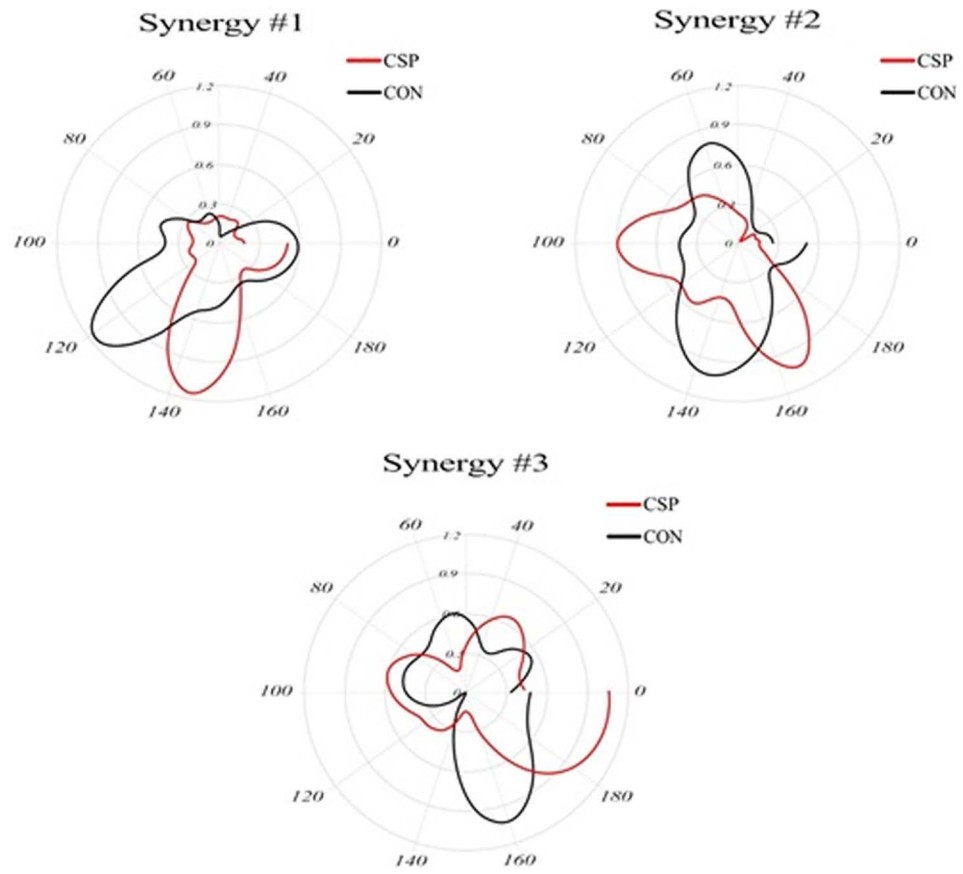

**Fig 4. Averaged synergy activation coefficients of Control Group (CON) and Subacromial Impingement Syndrome Group (SIS).** The left panel refers to Synergy #1, the central panel to Synergy #2, and the right panel to Synergy #3. The upper hemisphere of the graphs corresponds to the Forward phase (0–100% of the RRT cycle), while the lower hemisphere corresponds to the Reverse phase (100–0% of the RRT).

vectors demonstrated weaker to moderate correlations, with values of 0.20, 0.19, and 0.18 for synergies #1, #2, and #3, respectively, as presented in Table 4 and Fig 5.

Through an analysis of both synergy activation coefficients and muscle synergy vectors, distinct properties were identified for each muscle synergy. Synergy #1, primarily involves

**Table 4. Intra-group and inter-group similarity values (r) of muscle synergy vectors of the Control Group (CON) and Subacromial Impingement Syndrome Group (SIS).**

| Variable | Intra-Group | | | | Inter-Group |
|---|---|---|---|---|---|
| | CON (N = 12) | SIS (N = 12) | P-value[†] | ES | |
| | Mean ± SD | Mean ± SD | | | |
| # 1 | 0.21 ± 0.02 | 0.20 ± 0.02 | 0.582 | 0.014 | 0.20 ± 0.03 |
| # 2 | 0.20 ± 0.02 | 0.20 ± 0.03 | 0.765 | 0.004 | 0.19 ± 0.03 |
| # 3 | 0.20 ± 0.03 | 0.19 ± 0.04 | 0.684 | 0.008 | 0.18 ± 0.03 |

**Notes:** [†]statistically significant differences between the CON and SIS groups. **P-value,0.05 considered to be statistically significant.

Abbreviations: #1, Synergy #1; #2, Synergy #2; #3, Synergy #3; ES, effect size.

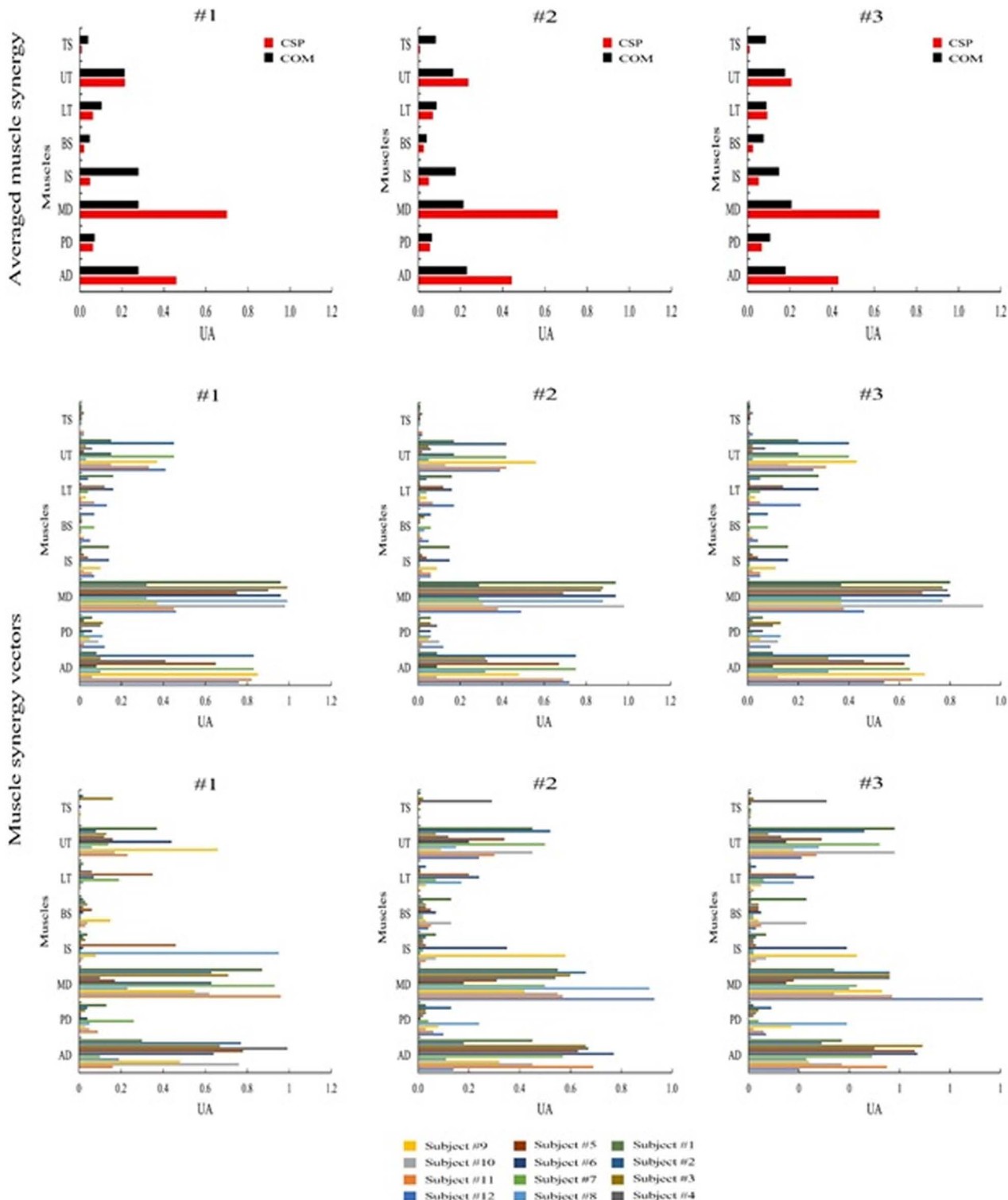

**Fig 5. Averaged muscle synergy vectors and muscle synergy vectors for all the subjects of the Control Group (CON) and Subacromial Impingement Syndrome Group (SIS) (UA–arbitrary units).** *Top panel*: averaged muscle synergy vectors between SIS and CON groups in the left panel refer to Synergy #1, the central panel to Synergy #2, and the right panel to Synergy #3. *Central and bottom panel*: muscle synergy vectors for all the subjects. Individual muscle weightings are depicted for each muscle within each synergy. The muscle synergy vectors have been normalized by their maximum under the synergy to which they belong. central panel refers to SIS, and the bottom panel to CON.

back muscles, including UT, LT, PD, IS, and TB. It becomes active during the reverse phase of the RRT, with peak activity occurring in the initial part of this phase (Fig 4 Central and bottom panel). Synergy #2, is predominantly active during the forward phase and, to a lesser extent, at the beginning of the reverse phase. It comprises activity in anterior upper-limb muscles, specifically BS, AD, and MD (Fig 4 Central and bottom panel). Synergy #3, is active from the middle of the forward phase to the start of the reverse phase. It mainly involves AD, MD, BS, TB, UT, and LT (Fig 5 Central and bottom panel).

Despite strong correlation values between groups (0.85 < r max < 0.92), additional temporal adjustments were observed. Within the intra-group analysis, significant backward shifts occurred in the PD (p < 0.001; effect size = 0.558) and TS (p < 0.027; effect size = 0.204). However, no significant delayed activation was evident in the inter-group comparison, as presented in Table 5 and Fig 6.

## Discussion

In this study, our primary aim is to examine the variances in movement variability and muscular coordination during the implementation of repetitive flexion and extension elbow movements in two groups of athletes with SIS and CON. After extracting the shoulder girdle muscle synergies, this study's results indicated three observable muscle synergy patterns in both groups. It was demonstrated that there was no significant difference in VAF between the SIS and CON groups of athletes for all three muscle synergies. Independent analysis of muscles showed that the VAF$_{muscle}$ significantly differed in the posterior deltoid and middle deltoid muscles in the SIS group compared to the CON group, whereas this value was lower significantly in the subscapular muscle in the SIS group. Additionally, no significant statistical

**Table 5. Intra-group and inter-group similarity values (r$_{max}$) and lag times (%) of synergy activation coefficients and individual EMG profiles of Control Group (CON); Subacromial Impingement Syndrome Group (SIS).**

| Variable | | Intra-Group | | | | | | Inter-Group | | | |
|---|---|---|---|---|---|---|---|---|---|---|---|
| | | CON (N = 12) | | SIS (N = 12) | | P-value† | ES | | | | |
| | | % Lag | r$_{max}$ | % Lag | r$_{max}$ | | | % Lag | r$_{max}$ | P-value† | ES |
| | | Mean ± SD | | | | | | Mean ± SD | | | |
| | | **Individual EMG profiles** | | | | | | | | | |
| Muscles | AD | 0.80 ± 0.31 | 85.32 ± 3.13 | 0.71 ± 0.24 | 85.14 ± 3.30 | 0.422 | 0.030 | −3.13 ± 3.20 | 90.38 ± 3.20 | 0.895 | 0.001 |
| | PD | 1.73 ± 1.34 | 85.56 ± 2.20 | −2.41 ± 4.60 | 85.53 ± 4.60 | 0.001** | 0.558 | −2.82 ± 3.53 | 91.47 ± 2.66 | 0.985 | 0.001 |
| | MD | 0.02 ± 3.19 | 90.02 ± 3.19 | 0.06 ± 2.60 | 90.06 ± 2.60 | 0.975 | 0.001 | 1.67 ± 2.85 | 93.59 ± 2.85 | 0.975 | 0.001 |
| | IS | 1.04 ± 3.57 | 86.04 ± 3.57 | 1.28 ± 1.22 | 88.53 ± 3.02 | 0.828 | 0.002 | −1.08 ± 3.47 | 93.42 ± 3.27 | 0.079 | 0.134 |
| | BS | 1.87 ± 2.20 | 89.21 ± 3.07 | 1.14 ± 2.20 | 88.22 ± 3.81 | 0.424 | 0.029 | 0.35 ± 3.42 | 90.27 ± 3.42 | 0.493 | 0.022 |
| | LT | 0.34 ± 2.10 | 85.50 ± 4.43 | 1.14 ± 2.22 | 87.21 ± 2.93 | 0.193 | 0.076 | −1.55 ± 3.06 | 89.73 ± 3.77 | 0.541 | 0.017 |
| | UT | −0.14 ± 2.09 | 91.94 ± 4.46 | 1.21 ± 2.81 | 92.62 ± 3.81 | 0.015 | 0.240 | 3.16 ± 3.59 | 95.65 ± 4.07 | 0.588 | 0.014 |
| | TS | −0.88 ± 1.52 | 85.87 ± 4.08 | 0.65 ± 1.64 | 88.41 ± 2.93 | 0.027** | 0.204 | −0.80 ± 3.04 | 92.51 ± 3.71 | 0.176 | 0.082 |
| **Synergy Activation Coefficients** | | | | | | | | | | | |
| # 1 | | 0.17 ± 0.39 | 82.24 ± 3.57 | 0.34 ± 0.62 | 81.06 ± 4.32 | 0.435 | 0.028 | −3.02 ± 2.71 | 86.92 ± 3.92 | 0.025** | 0.209 |
| # 2 | | -1.45 ± 1.60 | 86.21 ± 3.07 | −1.60 ± 2.50 | 85.22 ± 3.81 | 0.861 | 0.001 | −1.81 ± 2.21 | 87.99 ± 3.42 | 0.037** | 0.183 |
| # 3 | | -0.96 ± 0.88 | 81.20 ± 4.43 | −2.59 ± 2.52 | 82.91 ± 2.93 | 0.047** | 0.168 | −1.71 ± 3.70 | 82.33 ± 3.77 | 0.153 | 0.091 |

Abbreviations: TS: trapezius, IS: infraspinatus, AD: anterior deltoid, MD: middle deltoid, PD: posterior deltoid, BS, biceps brachii, TB, triceps medial head; ES, effect size. †Statistically significant differences between the CON and SIS groups. **P-value, 0.05 considered to be statistically significant.

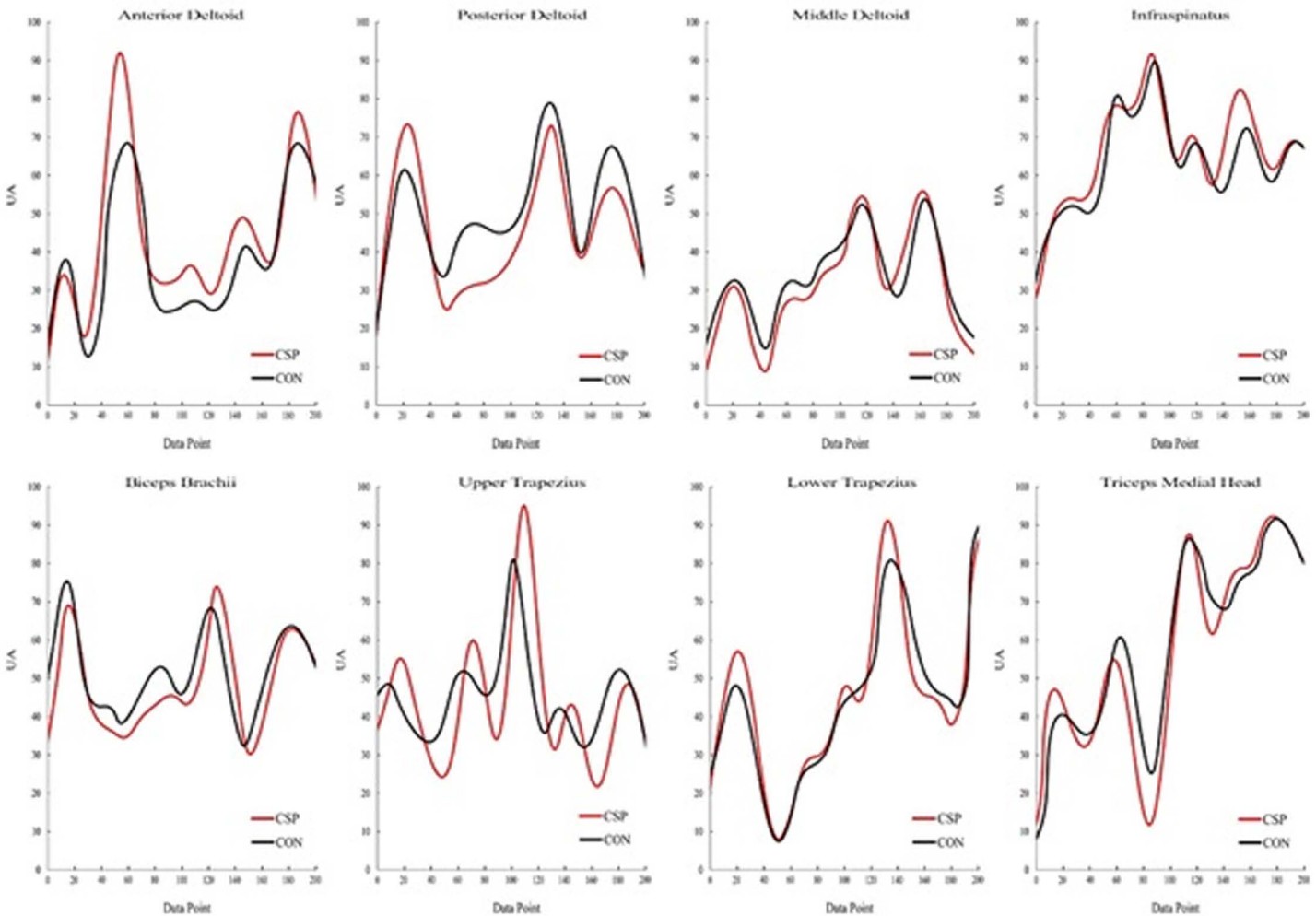

**Fig 6. Averaged EMG envelopes (UA—arbitrary units) from 8 muscles obtained in Subacromial Impingement Syndrome Group (SIS) and Control Group (CON) participants during the RRT (200-time points).**

differences were shown regarding Intra-Group Variability between the groups. However, the SIS and CON groups showed significant differences in Inter-Group Variability. Specifically, synergy 1 (shoulder joint posterior muscles and scapular posterior muscles) and synergy 2 (anterior shoulder joint muscles) were engaged earlier in the SIS group. In contrast, the CON group employed synergy 3 (arm muscles) earlier.

## Muscle synergies

For the SIS and CON groups, three muscle synergies were extricated based on the defined VAF criteria. Previous studies have also demonstrated muscle synergies are strong among athletes who require sports activities with repetitive movements, for example, seasoned cyclists [41], oarsmen [39], gymnasts [40], and athletes during paddling [39], and cycling [42]. We should acknowledge that the cross-validation results indicated that the SIS from the CON dataset reconstruction EMG patterns were significantly different, and the synergies number was similar. However, the presence of SIS causes an impact on muscular coordination and

movement variability during repetitive flexion and extension elbow movements in athletes significantly, which signifies that particular RRT results in varied neural strategies employed by the athletes. Moreover, have suggested that the shoulder girdle muscles in athletes with chronic shoulder pain are still able to generate effective movement patterns despite the presence of pain [43,44]. These results have important suggestions for understanding the musculoskeletal system's resilience and adaptability in the face of chronic pain [20,45].

Compared to the CON group, the SIS group exhibited significantly different VAF values in the posterior deltoid, middle deltoid, and subscapularis muscles. These differences indicate alterations in the activation patterns and coordination of these specific muscles in individuals with SIS. The lower VAF valuations observed in the subscapularis muscle in the SIS group may reflect a disruption in its normal recruitment pattern, potentially contributing to impaired shoulder stability.

## Intra-group variability

We observed that the synergy temporal component exhibited similar correlation values across different groups. Specifically, for the CON and SIS groups, the pairwise analysis did not reveal significant differences in the timing characteristics of the synergies. This consistency suggests that the temporal activation patterns of each synergy remained stable across subjects within each group. Additionally, the shape of the synergies showed little variability within each group. However, we did find notable differences in the correlation values between synergy #1 and #3 across the groups. Specifically, the CON group exhibited a higher degree of variability in this regard. In both the CON and SIS groups, the temporal activation patterns of muscle synergies remained remarkably consistent across subjects. This finding suggests that individuals within each group exhibited similar timing characteristics when activating these predefined motor patterns. Additionally, the shape of the synergies—representing coordinated muscle activations—showed little variability within each group. These consistent temporal and spatial patterns highlight the robustness of muscle synergies as fundamental building blocks for motor control [46–49].

## Inter-group variability

We found that during RRT, athletes with SIS had two peaks, synergy #1, in contrast to the CON group's synergy #1, which exhibited only a single peak. The observed difference could be attributed to the integration of synergy #2 and #3 in the SIS group. Specifically, this integration occurs at the transition between the final phase of reverse movement and the initial phase of forward movement. In the CON group, synergy #3 appears to involve elbow extension to reach the endpoint position primarily. However, in the SIS group, the third synergy pattern exhibits a second, increasing moment, with its peak aligning with the onset of activation of the first synergy. In the CON group, synergy #3 appears to involve elbow extension to reach the endpoint position primarily. However, in the SIS group, the third synergy pattern exhibits a second, increasing moment, with its peak aligning with the onset of activation of the first synergy. In contrast to the CON group, where synergy #1 exhibits no significant activation, the SIS group shows the presence of a peak in synergy #1. Considering this activation, it appears that synergy #1 in the CON group can be conceptually divided into both synergy #1 and the initial peak of synergy #3 observed in the SIS group. The observed fractionation leads to the inclusion of the PD in synergy #1 for the SIS group, whereas the MD (middle deltoid) and the TB predominantly contribute to synergy #3.

As anticipated, the muscle vectors exhibited low correlation values. However, there was an exception for synergy #1, which displayed less variability in its composition. In contrast, synergy #2 and #3 demonstrated weaker correlations. In the context of synergy #1, it appears

that the CON group relies on intrinsic synergies that have been previously utilized in similar motor tasks. Notably, the variation within both groups is approximately consistent with the variation observed between the groups. In the context of synergy #2 and #3, the observed weaker correlations could potentially be linked to the modulation of muscle synergies by the SIS. This is in line with previous studies' findings that specific strategies in response to individuals with chronic pain were developed [50].

Indeed, despite the study's limited sample size, most of the observed significant differences in muscle synergy components and individual EMG patterns exhibited a substantial effect size (Cohen's d greater than 0.5). These effects extend beyond the inherent variability in data related to muscle synergy extraction during the RRT [37,43,48,51,52]. The observed differences between the two groups in this study highlight the importance of further investigating adaptations in repetitive shoulder movements, especially among athletes. By including individuals with SIS in other tasks, we can assess how their coordination may evolve over time. Understanding the neural strategies behind complex movements and considering different phases of task processes will be crucial for promoting intermuscular adaptations. Additionally, incorporating kinematic and kinetic data could enhance our understanding of muscle activation during movement, complementing the results obtained from EMG analysis.

The results of this research offer clinical physiotherapists an important understanding of the changed neuromuscular coordination patterns in athletes experiencing SIS. By recognizing the particular muscles, including the posterior and middle deltoids, that show changed activation patterns, physiotherapists can create focused interventions to re-establish proper coordination and enhance functional performance. Rehabilitation programs may aim to retrain muscle synergies via exercises that improve intermuscular coordination, including proprioceptive and motor control activities [53]. Additionally, timely identification of changed synergy patterns might assist physiotherapists in applying preventative measures to minimize the likelihood of performance mistakes and additional injuries.

Although this article's results were interesting, the limitations of this research should be considered in order to generalize them. First, it should be noted that this study is a cross-sectional study. Therefore, its results cannot be attributed solely to shoulder pain in athletes. It should also be noted that motor strategies may change during exercise as athletes become fatigued. Therefore, it is recommended that this study be repeated when athletes are functionally fatigued. On the other hand, it should be noted that the movement evaluated in this laboratory study differs greatly from the movement of athletes during exercise training. Therefore, the results of the research may differ in real exercise conditions. Moreover, since simultaneous activation around the shoulder joint can provide valuable insights into joint stabilization and movement precision, consider incorporating muscle co-activation analysis into future studies.

## Conclusions

This study showed differences in muscular coordination and variability during RRT in athletes with and without SIS. Three different muscle synergy patterns were demonstrated in both groups, with notable differences in the VAF in selected muscles. It shows that athletes with SIS had significantly higher VAF in the posterior and middle deltoids and lower VAF in the subscapularis muscle than controls. Moreover, inter-group variability analysis disclosed that synergy #1 (posterior shoulder muscles) and synergy #2 (anterior shoulder muscles) were activated earlier in the SIS group. In contrast, the control group activated synergy #3 (arm muscles) earlier. It seems that timing and coordination changes in muscle activation may influence movement efficiency and increase the risk of performance errors.

## Supporting information

**S1 File. Supplementary file 1.**
(XLSX)

## Author contributions

**Conceptualization:** Rahman Sheikhhoseini, Sajjad Abdollahi, Mehrdad Anbarian.

**Data curation:** Rahman Sheikhhoseini, Sajjad Abdollahi, Mohammad Salsali, Mehrdad Anbarian.

**Formal analysis:** Rahman Sheikhhoseini, Sajjad Abdollahi, Mohammad Salsali, Mehrdad Anbarian, Trent M. Guess.

**Funding acquisition:** Rahman Sheikhhoseini, Sajjad Abdollahi.

**Investigation:** Rahman Sheikhhoseini, Sajjad Abdollahi.

**Methodology:** Rahman Sheikhhoseini, Sajjad Abdollahi, Mohammad Salsali, Mehrdad Anbarian, Trent M. Guess.

**Project administration:** Rahman Sheikhhoseini, Sajjad Abdollahi.

**Resources:** Rahman Sheikhhoseini, Sajjad Abdollahi.

**Software:** Rahman Sheikhhoseini, Sajjad Abdollahi, Mohammad Salsali.

**Supervision:** Rahman Sheikhhoseini, Sajjad Abdollahi, Mehrdad Anbarian, Trent M. Guess.

**Validation:** Rahman Sheikhhoseini, Sajjad Abdollahi, Mohammad Salsali.

**Visualization:** Rahman Sheikhhoseini, Sajjad Abdollahi, Mohammad Salsali, Trent M. Guess.

**Writing – original draft:** Rahman Sheikhhoseini, Sajjad Abdollahi, Mohammad Salsali, Mehrdad Anbarian, Trent M. Guess.

**Writing – review & editing:** Rahman Sheikhhoseini, Sajjad Abdollahi, Mohammad Salsali, Mehrdad Anbarian, Trent M. Guess.

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
