## [Decision Letter · Decision Letter 0]

2 Jan 2025

PONE-D-24-52678Coordination and Variability of Muscular Activation in Male Athletes with and Without Subacromial Impingement Syndrome: A Case-Control StudyPLOS ONE

Dear Dr. Sheikhhoseini,

Thank you for submitting your manuscript to PLOS ONE. After careful consideration, we feel that it has merit but does not fully meet PLOS ONE’s publication criteria as it currently stands. Therefore, we invite you to submit a revised version of the manuscript that addresses the points raised during the review process.

ACADEMIC EDITOR: Dear Author, please attend to all the comments provided by the reviewers and do the necessary corrections.

We look forward to receiving your revised manuscript.

Kind regards,

Zulkarnain Jaafar

Academic Editor

PLOS ONE

Journal Requirements:

“Iran National Science Foundation (INSF) under project no. 4013596”

Reviewers' comments:

Reviewer's Responses to Questions

**Comments to the Author**

1. Is the manuscript technically sound, and do the data support the conclusions?

Reviewer #1: Yes

Reviewer #2: Yes

2. Has the statistical analysis been performed appropriately and rigorously? 

Reviewer #1: Yes

Reviewer #2: Yes

3. Have the authors made all data underlying the findings in their manuscript fully available?

Reviewer #1: Yes

Reviewer #2: Yes

4. Is the manuscript presented in an intelligible fashion and written in standard English?

Reviewer #1: Yes

Reviewer #2: Yes

5. Review Comments to the Author

Reviewer #1: Dear Authors,

Special thanks for your manuscript submission and to the editors for providing the opportunity to review this work.

The study, titled Coordination and Variability of Muscular Activation in Male Athletes with and Without Subacromial Impingement Syndrome: A Case-Control Study,' aims to investigate and compare muscular coordination and variability during repetitive shoulder movements among athletes with and without SIS.

The manuscript is engaging, and here are some insights and suggestions:

- This research has the potential to interest the PLOS ONE readership but requires significant improvement.

- The abstract is unclear. Enhance it by clearly stating the study's aim, methodology, and conclusions for better clarity and comprehension.

- The authors investigate “the coordination and variability of muscular activation” using NNMF to extract muscle synergies. Additionally, since simultaneous activation around the shoulder joint can provide valuable insights into joint stabilization and movement precision, consider incorporating muscle co-activation analysis into the study.

- Use a single method (either tables or figures) to present results; avoid redundant representations, such as the overlap between Table 2 and Figure 3, or Table 3 and Figure 4.

- Revise the placement of Figure 6 and its legend to improve readability and alignment with the text.

- While the paper is well-illustrated and presented, it should be significantly reduced in length to focus on content directly relevant to the study's aim.

- The study's limitations are only briefly mentioned in the abstract. It is preferable to introduce a separate section for limitations and future work before the conclusions section.

- Although the conclusions section addresses the study's aim, it is too general. Improve it by incorporating the key findings and important results from the study.

Reviewer #2: 1) It is a interesting study however the authors need to elaborate why they have choosen the muscle synergies measurement and notmuscle activity?

2) what advatage it offers over the muscle acitvity?

3)the authors are advised to add comparision with muscle acitvities in SIS vide refercne "Sharma, S., Hussain, M. E., & Sharma, S. (2021). Effects of exercise therapy plus manual therapy on muscle activity, latency timing and SPADI score in shoulder impingement syndrome. Complementary Therapies in Clinical Practice, 44, 101390."

Phadke, V., & Ludewig, P. M. (2013). Study of the scapular muscle latency and deactivation time in people with and without shoulder impingement. Journal of Electromyography and Kinesiology, 23(2), 469-475.

4) the authors are requested to how weill the clinical physiotherpist use their findings to obtain good outcome

6. PLOS authors have the option to publish the peer review history of their article (what does this mean? ). If published, this will include your full peer review and any attached files.

**Do you want your identity to be public for this peer review?** For information about this choice, including consent withdrawal, please see our Privacy Policy .

Reviewer #1: **Yes: ** Abdel-Rahman Akl

Reviewer #2: No

---

## [Author Response · Author response to Decision Letter 0]

4 Jan 2025

Comments to the Author

1. Is the manuscript technically sound, and do the data support the conclusions?

Reviewer #1: Yes

Reviewer #2: Yes

Thank you for considering our work.

2. Has the statistical analysis been performed appropriately and rigorously?

Reviewer #1: Yes

Reviewer #2: Yes

Thank you for considering our work.

3. Have the authors made all data underlying the findings in their manuscript fully available?

Reviewer #1: Yes

Reviewer #2: Yes

Thank you for considering our work.

4. Is the manuscript presented in an intelligible fashion and written in standard English?

Reviewer #1: Yes

Reviewer #2: Yes

Thank you for considering our work.

5. Review Comments to the Author

Reviewer #1: Dear Authors,

Special thanks for your manuscript submission and to the editors for providing the opportunity to review this work.

The study, titled Coordination and Variability of Muscular Activation in Male Athletes with and Without Subacromial Impingement Syndrome: A Case-Control Study,' aims to investigate and compare muscular coordination and variability during repetitive shoulder movements among athletes with and without SIS.

The manuscript is engaging, and here are some insights and suggestions:

- This research has the potential to interest the PLOS ONE readership but requires significant improvement.

- The abstract is unclear. Enhance it by clearly stating the study's aim, methodology, and conclusions for better clarity and comprehension.

Thank you for your valuable comment. All parts were amended and highlighted.

- The authors investigate “the coordination and variability of muscular activation” using NNMF to extract muscle synergies. Additionally, since simultaneous activation around the shoulder joint can provide valuable insights into joint stabilization and movement precision, consider incorporating muscle co-activation analysis into the study.

Given that this study was not one of the primary objectives of this research and that the number of variables examined in this study is relatively large, this issue has not been examined. However, we have added this valuable point in the research limitations section.

- Use a single method (either tables or figures) to present results; avoid redundant representations, such as the overlap between Table 2 and Figure 3, or Table 3 and Figure 4.

Thank you for your valuable comment. Duplicates have been removed and necessary changes have been made to the text.

- Revise the placement of Figure 6 and its legend to improve readability and alignment with the text.

Thank you for your valuable comment. It is amended.

- While the paper is well-illustrated and presented, it should be significantly reduced in length to focus on content directly relevant to the study's aim.

Thank you for your valuable comment. Two figures were removed to shorten the text, and the authors also tried to summarize the text as much as possible. The word count of the article is based on the journal's standards. However, the large number of variables and the volume of results have increased the length of the article.

- The study's limitations are only briefly mentioned in the abstract. It is preferable to introduce a separate section for limitations and future work before the conclusions section.

Thank you for your valuable comment. The limitation section has been added to the text.

- Although the conclusions section addresses the study's aim, it is too general. Improve it by incorporating the key findings and important results from the study.

Thank you for your valuable comment. The conclusion changed.

Reviewer #2:

1) It is a interesting study however the authors need to elaborate why they have choosen the muscle synergies measurement and notmuscle activity?

Thank you for your valuable comment. We tried to add some text to introduction to address this issue.

2) what advatage it offers over the muscle acitvity?

Thank you for your valuable comment. We tried to add some text to introduction to address this issue.

3)the authors are advised to add comparision with muscle acitvities in SIS vide refercne "Sharma, S., Hussain, M. E., & Sharma, S. (2021). Effects of exercise therapy plus manual therapy on muscle activity, latency timing and SPADI score in shoulder impingement syndrome. Complementary Therapies in Clinical Practice, 44, 101390."

Thank you for your valuable comment. Added as reference.

Phadke, V., & Ludewig, P. M. (2013). Study of the scapular muscle latency and deactivation time in people with and without shoulder impingement. Journal of Electromyography and Kinesiology, 23(2), 469-475.

Thank you for your valuable comment. Added as reference.

4) the authors are requested to how weill the clinical physiotherpist use their findings to obtain good outcome

Thank you for your valuable comment. It is added to the discussion section, a paragraph before the limitations.

---

## [Decision Letter · Decision Letter 1]

27 Jan 2025

Coordination and Variability of Muscular Activation in Male Athletes with and Without Subacromial Impingement Syndrome: A Case-Control Study

PONE-D-24-52678R1

Dear Dr. Sheikhhoseini,

We’re pleased to inform you that your manuscript has been judged scientifically suitable for publication and will be formally accepted for publication once it meets all outstanding technical requirements.

Kind regards,

Zulkarnain Jaafar

Academic Editor

PLOS ONE

Additional Editor Comments (optional):

Reviewers' comments:

Reviewer's Responses to Questions

**Comments to the Author**

1. If the authors have adequately addressed your comments raised in a previous round of review and you feel that this manuscript is now acceptable for publication, you may indicate that here to bypass the “Comments to the Author” section, enter your conflict of interest statement in the “Confidential to Editor” section, and submit your "Accept" recommendation.

Reviewer #1: All comments have been addressed

Reviewer #2: All comments have been addressed

2. Is the manuscript technically sound, and do the data support the conclusions?

Reviewer #1: Yes

Reviewer #2: Yes

3. Has the statistical analysis been performed appropriately and rigorously? 

Reviewer #1: Yes

Reviewer #2: Yes

4. Have the authors made all data underlying the findings in their manuscript fully available?

Reviewer #1: Yes

Reviewer #2: Yes

5. Is the manuscript presented in an intelligible fashion and written in standard English?

Reviewer #1: Yes

Reviewer #2: Yes

6. Review Comments to the Author

Reviewer #1: It is preferable to introduce a separate section for limitations and future work before the conclusions section.

Reviewer #2: the authors have addressed all the concerns and it si good clinical study. in future if the authors study gets publsihed they should plan a DELPHI study

7. PLOS authors have the option to publish the peer review history of their article (what does this mean? ). If published, this will include your full peer review and any attached files.

**Do you want your identity to be public for this peer review?** For information about this choice, including consent withdrawal, please see our Privacy Policy .

Reviewer #1: **Yes: ** Abdel-Rahman Akl

Reviewer #2: No

---

## [Editor Report · Acceptance letter]

PONE-D-24-52678R1

PLOS ONE

Dear Dr. Sheikhhoseini,

I'm pleased to inform you that your manuscript has been deemed suitable for publication in PLOS ONE. Congratulations! Your manuscript is now being handed over to our production team.

Kind regards,

on behalf of

Dr. Zulkarnain Jaafar

Academic Editor

PLOS ONE